# Video Anomaly Detection with Contours - A Study

## Abstract

In Pose-based Video Anomaly Detection prior art is rooted on the assumption that abnormal events can be mostly regarded as a result of uncommon human behavior. Opposed to utilizing skeleton representations of humans, however, we investigate the potential of learning recurrent motion patterns of normal human behavior using 2D contours. Keeping all advantages of pose-based methods, such as increased object anonymization, the shift from human skeletons to contours is hypothesized to leave the opportunity to cover more object categories open for future research. We propose formulating the problem as a regression and a classification task, and additionally explore two distinct data representation techniques for contours. To further reduce the computational complexity of Pose-based Video Anomaly Detection solutions, all methods in this study are based on shallow Neural Networks from the field of Deep Learning, and evaluated on the three most prominent benchmark datasets within Video Anomaly Detection and their human-related counterparts, totaling six datasets. Our results indicate that this novel perspective on Pose-based Video Anomaly Detection marks a promising direction for future research.

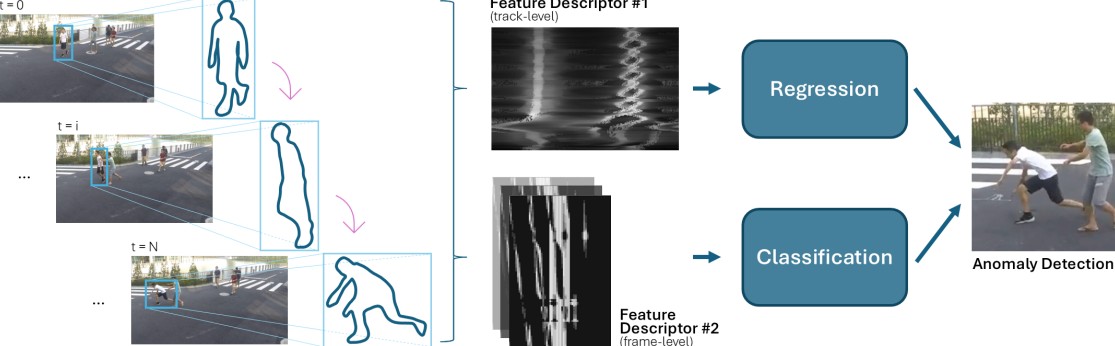

Figure 1: Overview of this study: Human contours are depicted by two distinct feature descriptors and analyzed by different regression and classification models to detect anomalous behavior in video.

## 1 Introduction

The differentiation between common and uncommon events in video footage by means of Artificial Intelligence (AI) is typically referred to as Video Anomaly Detection (VAD). Within the context of this particular Computer Vision (CV) task, the overall aim is to develop a statistical model capable of automatically distinguishing between more and less frequent events observed in a scene. The usual naming convention in existing literature is to speak of these events as *normal* and *abnormal/anomalous*, respectively.

Based on the assumption that unusual events most commonly result from abnormal interactions between objects and/or humans, many existing solution approaches within VAD employ an object-centric perspective on this task (Ionescu et al., 2019; Ouyang & Sanchez, 2021; Georgescu et al., 2021b; Doshi & Yilmaz, 2023; Siemon et al., 2024). The emergence of vision transformer-based network architectures in Deep Learning (DL) (Vaswani et al., 2017), enabled the excelling success of more basic CV tasks like object detection and

segmentation. With these pretrained models, the most accurately performing VAD solutions to date extract the required regions of interest from the given video frame, and process these patches on pixel-level using regression and/or classification approaches. During the last stage of those VAD pipelines, each region of interest is labeled with a (probability) score which indicates how abnormal it is.

Because the demand of computational resources supporting pixel-level VAD pipelines is usually considerably high, a trend towards object-centric approaches based on different levels of object abstractions has advanced in most recent years. Guided by the hypothesis that abnormal interactions may be predominantly traced back to humans acting as either catalysts or signifiers, the area of so-called Pose-based VAD (PAD) has been attracting a lot of attention. These human-centric approaches which operate on different levels of appearance abstractions such as skeletons, exhibit privacy-preserving, explainability, transferability, and low resource demand characteristics.

In contrast to pixel-based approaches, methods based on such object abstractions/simplifications have also shown to be transferable across the imaging spectrum where they outperform prior pixel-based art (Dueholm et al., 2024). Further, these findings pave the way to include the synthetic imaging modality in VAD-related experimental setups. As such, it bears the promising potential of providing an unlimited amount of video footage showing artificially generated anomalies, including highly accurate object annotations like bounding boxes, segmentation masks, and skeletons. This opportunity enables new approaches within the field which can now tackle the real-world open-set character of VAD.

Inspired by this and the latest trends within the emerging field of PAD, we thus present a study of utilizing human contours/silhouettes for PAD. By including the benchmark dataset UBnormal (Acsintoae et al., 2022) into the pool of datasets which we evaluate our approaches on, we prove that contours represent a level of appearance abstraction that is agnostic to real and synthetic image data in the RGB domain. Additionally, opposed to skeletons, this approach leaves the opportunity to be expanded to other object categories open for future research.

The contributions proposed in this paper are thus as follows:

- A study of the potential of contours as sufficient appearance abstractions of humans for PAD, based on regression and classification approaches from the field of DL.

- The analysis of two data representation techniques for contours, one naive, and one based on the concept of Shape Contexts.

- The preservation of increased object anonymization with reduced computational complexity.

- Several baseline models for working on the proposed data representation, giving an understanding of expected performance and to be used as easy comparisons by future researchers.

## 2 Related Work

**Video Anomaly Detection** Approaches within the field of VAD mostly fall into two categories: object-centric/patch-based (Ionescu et al., 2019; Ramachandra & Jones, 2020; Ouyang & Sanchez, 2021; Georgescu et al., 2021b; Doshi & Yilmaz, 2023; Singh et al., 2023; Siemon et al., 2024; Astrid et al., 2021), and frame-level (Liu et al., 2018; Park et al., 2020; Liu et al., 2021; Nikolov, 2023; Zhao et al., 2022) ones. Regardless, however, of whether these approaches operate on a local neighborhood of pixels or on the entire frame, the majority of them adopts unsupervised learning techniques from Machine Learning (ML) and/or DL to learn/memorize conspicuous patterns in video footage featuring exclusively common/normal events. This is attributed to the fact that anomalous incidents are very scarce in comparison to their normal counterparts, and because they substantially represent the *unexpected*. Formulating the task of VAD as a supervised learning problem may therefore impose a big challenge.

Within the pool of unsupervised/self-supervised learning methods for VAD, the most prominent areas covered are reconstruction-based, distance-based, or probabilistic in nature (Ramachandra et al., 2020). While some operate on the raw image itself, or local patches thereof, others rely on processing latent features extracted

using Neural Networks (NN). Only a few are based on the detection of semantically differentiable anomalous (inter-)actions (Doshi & Yilmaz, 2023). All in all, most successful prior art focuses on formulating the problem as a synergy of varying proxy tasks. As such, it is most commonly split into modules focusing on the detection of appearance-, and motion-based anomalies. Examples of such works include Park et al. (2020); Georgescu et al. (2021a); Singh et al. (2023). In Park et al. (2020), the authors perform frame-based reconstruction and prediction using a Convolutional Auto-Encoder (CAE) architecture augmented by an additional memory module to memorize feature representations of frames in latent space. The contribution of Georgescu et al. (2021a), on the other hand, is a hybrid model based on joint learning of four object-centric, self-supervised detection, prediction, and knowledge distillation tasks. These tasks are primarily focusing on the arrow of time, motion (artificially infused irregularity, and prediction), and the appearance of objects. The most successful approach achieving the highest accuracy scores to date is presented by Singh et al. (2023). In their 3D patch-based approach, the authors distinguish between five hand-crafted attributes depicting appearance, motion (angle and magnitude), and background characteristics (stationary pixels and a binary class). Learning a set of normal latent features representing these attributes using a greedy ML approach, allowed them to present an explainable VAD solution proposal that exceeds the results of prior art by a large margin.

**Pose-based Video Anomaly Detection**  Based on the assumption that the main source of abnormal events are humans, which is also a common argument to use pre-trained closed-set object detectors for object-centric VAD, the field of PAD has started to attract a lot of attention in most recent years. Even though the authors of publications within PAD admit that the superiority in performance accuracy lies with pixel-based VAD methods, they also underline the distinctive strengths of their approaches. Because they mostly operate on skeletons of humans to detect abnormal human behavior (excluding all other object classes), they are said to require fewer computational resources while respecting the privacy of the recorded individuals. Prominent works within this subfield of VAD include Morais et al. (2019); Markovitz et al. (2020); Hirschorn & Avidan (2023); Noghre et al. (2024). Morais et al. (2019), for example, learn regular motion patterns of human trajectories by means of a joint learning framework that processes dynamic local and global skeleton features. This learning framework is composed of two recurrent encoder-decoder network branches. Markovitz et al. (2020), on the other hand, propose a spatiotemporal graph CAE to project sequences of human skeletons into latent feature vectors which are further clustered into most prominent normal action types. The differentiation between normal and abnormal human (inter)actions is then determined utilizing a Dirichlet process mixture model. Similarly to Markovitz et al. (2020), Hirschorn & Avidan (2023) represent human skeleton trajectories employing spatio-temporal graphs, mapping these motion abstractions, however, into a latent Gaussian distribution instead. Lastly, Noghre et al. (2024) present an exploratory study of utilizing Variational Auto-Encoder (VAE) architectures to predict skeleton-based trajectories of human motion. In their approach, the authors propose a joint learning framework that operates in two parallel phases by reconstructing individual poses and the trajectory they represent.

Along the same line of argumentation as used in PAD, Siemon et al. (2024) recently presented their object-centric approach for VAD in which the authors solely rely on the analysis of high-level bounding box attributes within a probabilistic ML framework. Inspired by the reported results in combination with the emergence of PAD, this work explores the potential of utilizing human contours for the detection of abnormal events in video. This can be seen as an extension of the work presented by Siemon et al. (2024) given that it expands to capture behavior-based anomalies which do not only rely on abnormal appearance and/or motion attributes.

## 3  Methodology

### 3.1  Data Pre-Processing

In order to extract the contours of all humans which appear in a given scene, all videos are preprocessed using two pre-trained DL modules: a Multi-Object Tracker (MOT) and a Semantic Segmentation model. The former is employed by means of a BoT-SORT (Aharon et al., 2022) instance using YOLOv7 as the object detection backbone, pretrained on MS-COCO (Lin et al., 2014). Additionally, we enable the re-identification module (pretrained on MOT-17 (Milan et al., 2016)) to obtain more accurate tracking results.

The segmentation module, on the other hand, is implemented by means of the Vision Transformer (ViT) called Segment Anything (SAM) (Kirillov et al., 2023). We use the pretrained network model ViT-H, and provide the previously extracted bounding boxes as visual prompts for SAM to generate the corresponding object masks.

Last but not least, we develop a custom algorithm to extract the contour of a given mask traced with a single stroke. Each candidate contour is represented as the boundary points of a 2D shape excluding potential holes. The final qualification requirements for a candidate contour to be valid are as follows: (1) The Euclidean distance between the first and last point on the traced contours must not be greater than three pixels, and (2) each contour, depending on the data representation type, must be composed of at least 100/256 elements. More details about this follow in the subsequent section. All objects, represented in polar space via $(r, \theta)$ pairs, are normalized on video-level with respect to the largest object of the same class encountered in the given scene. On track-level, we disregard all sequences which are composed of five or less contours.

### 3.2 Data Representations

Throughout the course of experiments in which we apply various problem modeling perspectives, we exploit two distinct data representation techniques. The first one depicts a custom and more naive representation of contours, further referred to as *Radii-based Feature Descriptor*. The second one, on the other hand, has been introduced by Belongie et al. (2000) and represents a well-established descriptor for object shape matching in 2D which has also been extended to the problem of object detection, and 3D (Körtgen et al., 2003). We motivate the choice of these two feature descriptors for the extracted contours based on the distinctive properties they entail. Apart from both approaches being sensitive to rotational changes of the given object, they also facilitate different sets of experiments. A more detailed comparison is given in Table 1 followed by an illustration of both processes in Figure 2.

Table 1: Comparison of the two employed feature representation techniques

| Name | **Radii-based Feature Descriptor** | **Shape Contexts** |
| --- | --- | --- |
| Motivation | Sufficient reduction from 2D to 1D | Enables shape matching in 2D |
| Requires $N$ uniformly sampled points | 256 | 100 |
| Implementation | **One-to-one** representation of points in polar space such that each point is mapped to its absolute distance to the object's center of mass which serves as the origin. These 1D descriptors are stacked vertically to obtain track-wise motion features. | **One-to-many** representation of points treating every sampled point as the origin of a local coordinate system in polar space. $(r, \theta)$ pairs of all remaining points are discretized into a $(5 \times 12)$ histogram, as per recommendation of Belongie et al. (2000). As a results, we obtain a $(100 \times 60)$ feature descriptor for each contour which is optionally flattened and stacked (vertically) to produce track-wise descriptors. |
| Primarily for ... | Regression | Classification |

In the next subsections, we will discuss the regression and classification ways of utilizing the extracted data representations. We propose several architectures to utilize these representations, which will be later tested in Section 4.

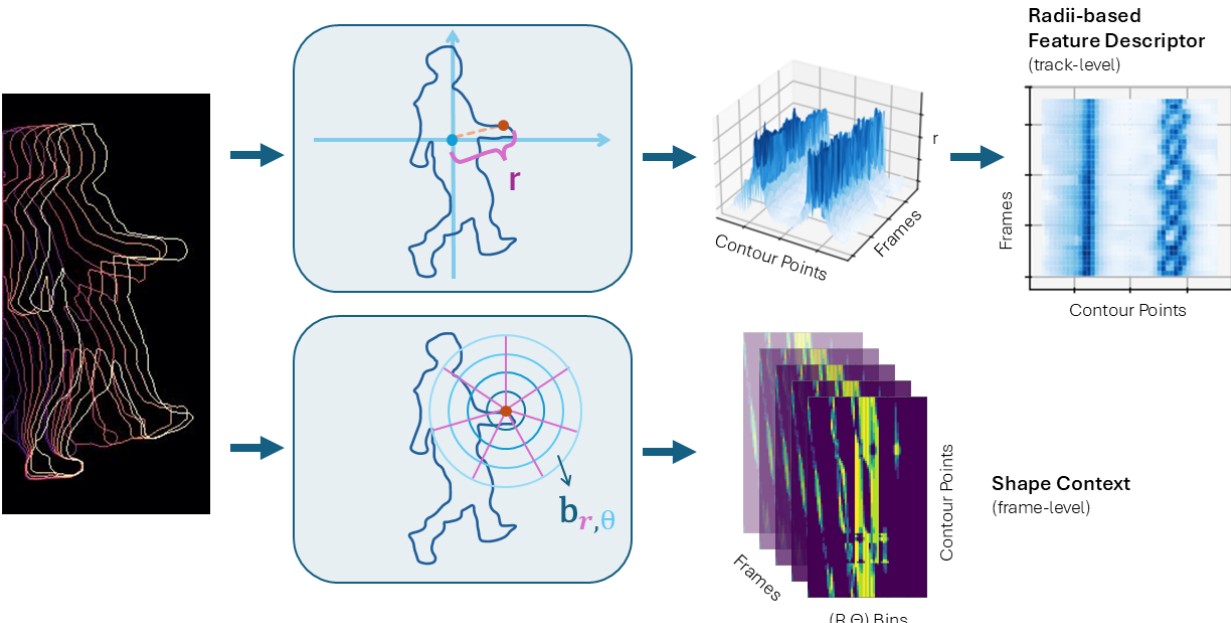

Figure 2: Data generation pipeline for both feature descriptors

## 3.3 Following the Regression Way

We test different shallow NN architectures to reconstruct the proposed radii-based feature descriptor images for each captured pedestrian track. As the input images have varying heights based on the length of the tracked trajectories, the proposed architectures are thus modeled to ingest them in different ways. By applying vertical stretching and/or stacking of the naive feature descriptors, we hope to minimize the loss of temporal information encoded within them. As a consequence, the proposed models come at different levels of complexity: First, we present the use of a traditional Variational Auto-Encoder (VAE) based on convolutional layers as a baseline which takes the radii-based feature descriptors and resizes them to quadratic images. Second, we propose the use of a simplified Linear Auto-Encoder (LAE) that only contains linear layers in its architecture processing the track descriptors in the same way as the VAE. Third, we propose to combine all radii-based feature descriptor data into a tabular form and forgo separation into images without employing any resizing. With this data representation, we further simplify the problem and utilize a Tabular linear Auto-Encoder (TAE). Lastly, we propose a sequential modeling approach to the problem of utilizing the radii-based feature descriptors, using a Regression-based Recurrent Neural Network (R-RNN). In this approach, we use individual shapes as track-wise sequential inputs to the network to predict the subsequent shape within a track. Each proposed architecture along with its motivational background is discussed in the following subsections.

### 3.3.1 Variational Auto-Encoder (VAE)

The first model we use is a baseline convolutional VAE, widely used in anomaly detection and reconstruction tasks (An & Cho, 2015; Pol et al., 2019; Lin et al., 2023). These Auto-Encoder (AE) models are commonly used for their better representation of latent space, which can be more easily separated than other types of convolutional AEs (Klys et al., 2018). The radii-based feature descriptor images we generate, can have very similar visual shapes and repeated textures, so our hypothesis with utilizing a VAE model is that it can differentiate the smaller details and changes in the images caused by the different trajectories captured in them. We utilize a simple VAE, shown in Figure 3, consisting of an encoder and decoder with four convolutional layers each. Together with two fully connected layers just before the latent space to represent the mean and the logarithm of the variance of the latent Gaussian distribution. To train the model, we utilize a combination of Mean Squared Error (MSE) and Kullback-Leibler divergence loss (Kingma, 2013). We train

it for 100 epochs, using a batch size of 16, together with an Adam optimizer and a learning rate of $1e-4$. We select these parameters after doing a parameter search and choosing the best possible combination. As the VAE takes in square images, we reshape the image of each radii-based feature descriptor so that the temporal dimension (y-axis) is of the same size as the number of features resulting in $(256 \times 256)$ images.

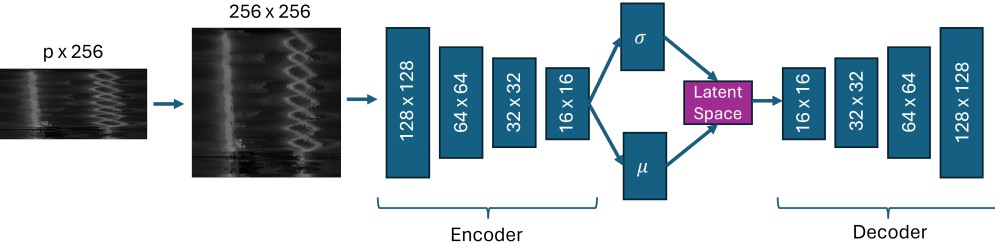

Figure 3: Architecture of the Variational Auto-Encoder (VAE) with four convolutional layers, serving as our baseline. The radii-based feature descriptor images are resized into quadratic form before training.

### 3.3.2 Linear Auto-Encoder (LAE)

One problem that can arise from utilizing a convolutional VAE, is that the architecture can become too complex for the input data which can cause overfitting and a lack of generalization. The radii-based feature descriptor images are created from outline data points, which can be seen as a naive and highly simplified data representation of object contours. To verify if this is the case, we build a simple feedforward AE architecture that has an encoder and decoder part built only from linear fully connected layers. A ReLu layer serves for regularization in the encoder and decoder including a Sigmoid layer after the final one of the decoder. We explore different sizes of fully connected LAEs and see that the best results are provided by a two-layer one. We train the LAE for 100 epochs, with a batch size of 8, an Adam optimizer, and a learning rate of $1e-4$. We visualize the architecture in Figure 4.

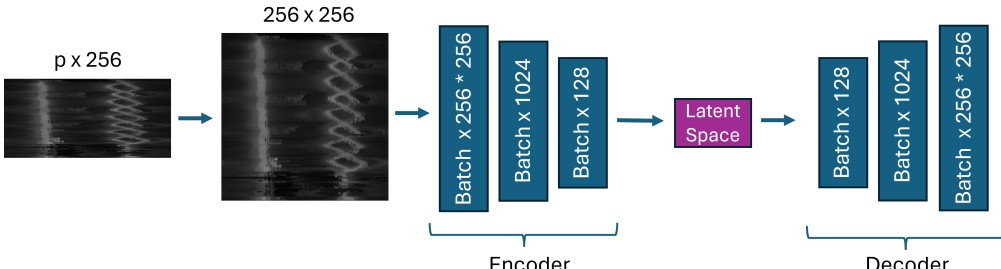

Figure 4: Architecture of the feedforward Linear Auto-Encoder (LAE) with two fully connected layers. The radii-based feature descriptor images are resized into quadratic form before training.

### 3.3.3 Tabular Auto-Encoder (TAE)

In the last NN which is inspired by the basic AE architectures, the proposed radii-based feature descriptor training is further reduced by looking at it as a time-series type of problem (Kieu et al., 2019). To do this, we combine all the images representing all tracked shapes into one long tabular form. In this way, we remove the need for reshaping the temporal dimension of the data aiming at better preserving the time dimensionality. Taking inspiration from other tabular data time-series research (Liguori et al., 2021; Tavakoli et al., 2020; Fan et al., 2018), we visualize our proposed TAE architecture in Figure 5. The encoder and decoder are each built of four fully connected linear layers, with ReLu regularization, together with a Sigmoid layer at the end of the decoder. The model is trained with a batch size of 16, using a MSE loss for 50 epochs. An Adam optimizer with a learning rate of $2e-4$ is used for faster convergence. Similarly as before, these values are selected by exploring the parameter space and selecting the ones that provide the best results.

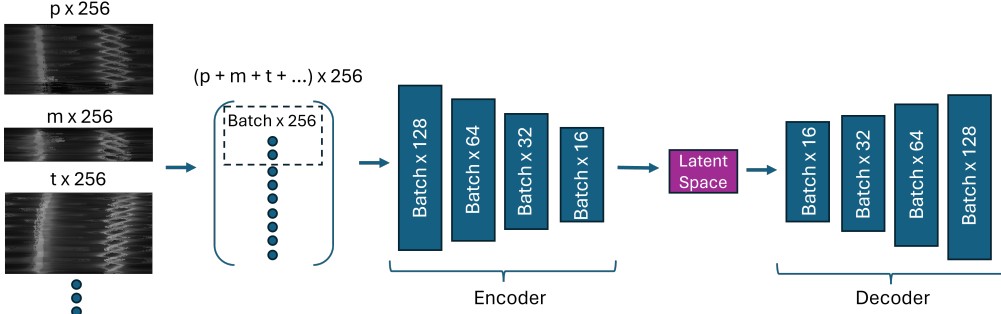

Figure 5: Architecture of the linear Tabular Auto-Encoder (TAE) with four fully connected layers. The radii-based feature descriptor images are transformed into tabular representation before training.

### 3.3.4 (Regression) Recurrent Neural Network (R-RNN)

In the last setup of modeling techniques which approach the problem as a regression task, the goal is to perform future contour prediction of humans. The main motivation behind this perspective, is to explore the potential of sequence modeling approaches to capture potentially recurrent patterns as well as disruptions in a sequence of non-rigid deformations of human motion.

Based on three consecutive object contours appearing in three consecutive frames, the aim is thus to predict the fourth one. Representing each contour by means of the radii-based feature descriptors described in Section 3.2, allows us to treat each descriptor as a flattened signal to be regressed/predicted based on knowledge of its appearance at preceding time steps. For this purpose, a custom Recurrent Neural Network (RNN) architecture, where weights are optimized using Adam and the loss is calculated as the MSE of the prediction with respect to the actual contour, is implemented. Based on an extensive grid search, the most promising network topology is chosen to consist of a single layer with 8 hidden neurons, and a the Hyperbolic Tangent as the activation function, as visualized in Figure 6. The network is trained for 200 epochs at a learning rate of $1e - 5$, and follows the many-to-one paradigm in common sequence modeling such that the input and output size of the network is equal to the length of a contour, i.e., 256.

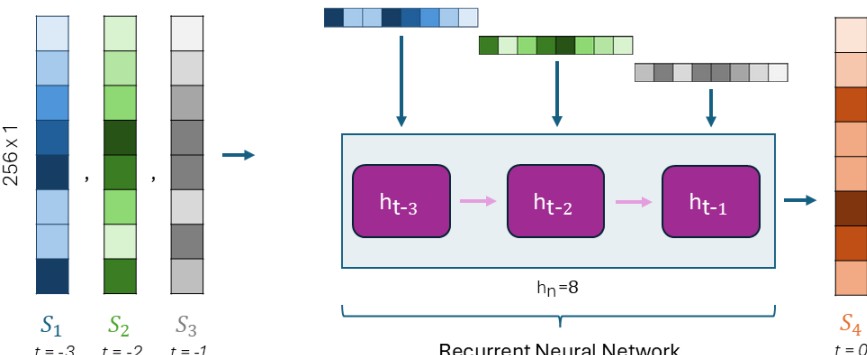

Figure 6: Architecture of the Recurrent Neural Network for Future Contour Prediction (R-RNN) with one hidden layer of 8 neurons. During training, the preceding object contours, represented by the naive radii-based feature descriptor, are fed into the network one time step at the time.

### 3.4 Following the Classification Way

Formulating the task of contour-based VAD as a classification problem constitutes an extension to the experiment described in Section 3.3.4 and may therefore also be called *cluster-based* future contour prediction of objects. It represents the concluding approach of our study and aims at extending the exploration of sequence modeling approaches from a regression to a classification perspective in order to distinguish

between normal and abnormal patterns in human motion. All in all, this approach is composed of three steps which are detailed in the following subsections.

### 3.4.1 Shape Clustering

Formulating the underlying problem as a classification task, is facilitated by the exploitation of Shape Contexts for the purpose of shape matching. Mapping entire contours composed of 6 000 numeric elements in an array to a single category is accomplished via their discretization into most representative clusters governing the training data. Clustering using the $\chi^2$ distance metric is hereby performed using the concept of Hierarchical Clustering (HC).

Due to the computational demands of this clustering algorithm, we develop a two-step self-supervised labeling approach to annotate all contours in the training data: First, we randomly select 10 000 training contour samples and generate their respective Ground Truth (GT) labels/categories using HC. Second, we train a multi-class Support Vector Machine (SVM) to produce the labels for the remaining contours in the training set. The best hyperparameters for the SVM are determined using $k = 5$ stratified randomized folds keeping 20% within each fold for validation. This yields $C = 1e5$ and $\gamma = 1e-3$. We use the same cross-validation approach to obtain the most accurate classification model for further labeling. The output of the annotation process is additionally validated by comparing the overall distribution of samples across clusters with the one of the GT labels.

### 3.4.2 Shape Classification

Once the tracks are discretized into sequences of most prominent contour cluster representations, similarly as in Section 3.3.4, we perform the actual future contour prediction using a many-to-one RNN model. An extensive grid search over a subspace of the set of possible hyperparameters reveals the following network topology to perform best in combination with the Adam optimizer and the Categorical Cross-Entropy loss: A single hidden layer with 8 neurons and the Hyperbolic Tangent as the activation function. The network is trained for 200 epochs at a learning rate of $1e-3$. We visualize the network in Figure 7.

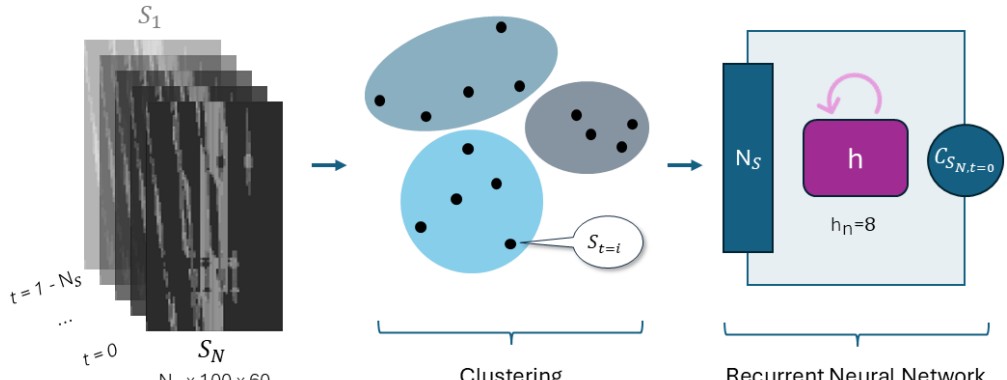

Figure 7: Architecture of the Recurrent Neural Network for Cluster-based Future Contour Prediction (C-RNN) with one hidden layer comprising 8 neurons. Each contour $S_i$ of a track comprising $N_S + 1$ frames, is represented via its Shape Context. During training, for every track, all $N_S$ contours preceding the current frame, discretized into most prominent clusters, are fed into the network one time step at the time. Lastly, the RNN outputs the cluster label $C_i$ depicting the most likely cluster for contour $S_N$ of the current frame.

In order to further account for the skewed distribution of the lengths of sequences, and the comparably smaller amount of data compared to the regression task, we additionally implement a data augmentation approach which is based on slicing approximately 80% of sequences selected at random in every epoch into smaller chunks of a dynamically determined length. The length of these slices, is set to fall within the range [min_range, sequence_length], where min_range $= 3$. Random shuffling of these slices further yields a more balanced prediction performance of the model across different positions within a sequence.

Performing a grid search using three different numbers of clusters $(10, 30, 50)$, we find that 30 in combination with 8 neurons in the hidden layer of the RNN, yields the most accurate performance results across all benchmark datasets. Because we dynamically discard those clusters which contain less than a relative percentage of samples, the actual amount of clusters $N_C$ falls in the interval $N_C \in [20, 30)$.

### 3.4.3 Novelty Detection

The last step in this sequence classification pipeline is designed to detect out-of-distribution samples which are too far from the clustered contours present in the training set. Given that anomalous motion can on one hand be caused by an anomalous flow of common poses, it can also be attributed to the appearance of novel poses which cannot be found in the training data. To address the latter, we complete our pipeline with a novelty detection module in the form of a One-Class SVM. This model is trained in an unsupervised fashion on the same subset of 10 000 training contours as the classification SVM used for labeling purposes in Section 3.4.1 using the non-linear RBF kernel and setting the parameter $\gamma = 1e - 3$.

Lastly, we describe every track and its corresponding sequence of $N$ contour clusters as an array of $N - 1$ transitional probabilities which are the result of multiplying the probability score from the C-RNN with the probabilistic proximity of the contour itself to the training data distribution.

## 4 Experimental Results

### 4.1 Datasets

We evaluate our unsupervised learning methods presented in Section 3 on a total of six benchmark datasets: CUHK Avenue (Lu et al., 2013), HR-Avenue (Morais et al., 2019), ShanghaiTech (Liu et al., 2018), HR-ShanghaiTech (Morais et al., 2019), UBnormal (Acsintoae et al., 2022) and HR-UBnormal (Flaborea et al., 2024). Representing a very broad range of background, recording angles and anomaly characteristics, details of these prominent VAD datasets are shortly described in the proceeding paragraphs.

**CUHK Avenue**  As a single-scene benchmark dataset for VAD, CUHK Avenue consists of 16 training and 21 test videos, with 15 328 and 15 324 frames, respectively. This dataset features anomalous events that either originate from the unique appearance or motion characteristics of the given object. Abnormal interactions of humans with objects are introduced to the test set utilizing the deliberate performance of unusual behavior. As such, these anomalies are rather simple in nature.

**ShanghaiTech**  ShanghaiTech is one of the most sophisticated VAD datasets released to date. Composed of recordings from 13 scenes, this multi-scene benchmark dataset contains 130 anomalous events with 274 515 training and 42 883 test frames. Similarly, as in CUHK Avenue, the majority of abnormal activities are performed by actors. In contrast to CUHK Avenue, however, this dataset also depicts anomalous interactions between two or more humans.

**UBnormal**  The first open-set VAD benchmark dataset was introduced by Acsintoae et al. (2022) and is fully synthetic. UBnormal is composed of 147 887 normal and 89 015 abnormal video frames, distributed across 543 videos. It is the first VAD dataset that looks at anomalous events from a real-world application perspective. To this end, this fully annotated dataset which can be used for both unsupervised and supervised training approaches, follows the open-set convention of anomalies by introducing anomalies during testing that are not included in the training videos. We note that for the sake of following the same protocol for our experimental setups, we only use the unsupervised training and test splits for evaluation.

**HR-Avenue, HR-ShanghaiTech and HR-UBnormal**  With the emergence of the field of PAD, the need of benchmark datasets depicting solely human-related anomalies emerged. For this reason, Morais et al. (2019) and Flaborea et al. (2024) suggested the removal of those few frames/videos which do not contain humans as catalysts/signifiers of anomalous events from CUHK Avenue, ShanghaiTech and UBnormal, introducing *HR-Avenue*, *HR-ShanghaiTech* and *HR-UBnormal*, respectively.

### 4.2 Evaluation Metrics

The performance of our methods is assessed by an object-centric evaluation protocol calculating the following accuracy metrics: Frame-level AUC, Region-Based Detection Criterion (RBDC), and Track-Based Detection Criterion (TBDC). In order to calculate the two latter for CUHK Avenue and ShanghaiTech based on the implementation published by Georgescu et al. (2021b) alongside their paper, we use the ground truth annotations on object- and track-level provided by Ramachandra & Jones (2020). With respect to the human-related counterparts of these two datasets, we follow the common convention in PAD as suggested by Morais et al. (2019), and exclude those frames from the respective datasets which either depict non-human anomalies or those in which the anomalous person is too occluded to provide sufficient visual means for the extraction of a statistically meaningful contour. The same applies to UBnormal and HR-UBnormal. Similarly to Georgescu et al. (2021b), we apply a Gaussian filter across the temporal axis to level the obtained frame-level AUC scores.

### 4.3 Results

We report the results of our experiments in Tables 2 and 3 by dividing them into VAD- and PAD-oriented sets, respectively. Both sets are further discussed in more details in the respective paragraphs below.

**VAD Benchmark Datasets (Table 2)** Considering the results we obtained based on our proposed regression approaches, we find that the TAE yields the best results on average across all VAD benchmark datasets compared to our remaining models. At the same time, we can observe that the classification model C-RNN performs worst overall. We assume, that this might be attributed to the fact that the underlying data discretization process is too drastic, leading to the loss of valuable details that can not be balanced with the proposed data augmentation step.

Comparing the different AE architectures, it can be seen that the linear models perform better on all datasets than the variational baseline. The closest gap between state-of-the-art AUC and TBDC scores is hereby achieved by the LAE on UBnormal where we perform 0.50% and 1.75% lower, respectively.

We experience the lowest performance in contrast to prior art on the benchmark dataset ShanghaiTech by a significant margin on all metrics. This is most likely due to the large variance in distribution of contours that differentiates this dataset from others. Compared to CUHK Avenue, for example, the recorded people are walking in a diverse range of directions. When it comes to UBnormal, on the other hand, the recording perspectives/angle is observed to change more significantly, which can affect contours much more than skeletons.

**PAD Benchmark Datasets (Table 3)** Similar to the observations based on the results reported in Table 2, the linear AE architectures (LAE and TAE) perform best across all experiments we conducted, with a significant improvement over our baseline model (VAE). Taking a close look at the results we achieved on the dataset HR-UBnormal additionally reveals that all our AEs in fact surpass prior art by at least 3% in terms of frame-level AUC setting a new state-of-the-art score. What we also observe on both UBnormal and HR-UBnormal, is that the classification RNN (C-RNN) yields more accurate results than the regression RNN (R-RNN). This stands in contrast to what can be noticed with respect to the remaining datasets. The lowest performance remains to be seen on HR-ShanghaiTech which we explain based on the same reasoning used before.

Overall, a slightly improved performance becomes apparent when comparing results on VAD- and PAD-based benchmark datasets. This may indicate that the influence of frames holding anomalous objects within these datasets remains potentially small.

## 5 Conclusion

In this work, we presented an exploratory study to use human silhouettes/contours in 2D to perform human-centric VAD. Similar to skeleton-based methods in PAD, we followed the motivation to develop AI models for

Table 2: Frame-level AUC, RBDC and TBDC scores achieved on the datasets CUHK Avenue, ShanghaiTech and UBnormal. We compare the results of our unsupervised approaches against other object-/human-centric, and patch-based approaches from VAD and PAD. Our regression- and classification-based models are marked with an (R) and (C), respectively. We highlight the best results per metric and within each category (VAD/PAD/Ours) with bold font.

| | Method | CUHK Avenue | | ShanghaiTech | | UBnormal | |
|---|---|---|---|---|---|---|---|
| | | AUC | RBDC/TBDC | AUC | RBDC/TBDC | AUC | RBDC/TBDC |
| VAD | Georgescu et al. (2021b) | **92.30** | 65.05/66.85 | **82.70** | 41.34/78.79 | **59.30** | **21.91/53.44** |
| | Singh et al. (2023) | 86.02 | **68.20/87.56** | 76.63 | **59.21/89.44** | - | -/- |
| PAD | Flaborea et al. (2023) | - | -/- | - | -/- | 68.30 | -/- |
| | Hirschorn & Avidan (2023) | - | -/- | **85.90** | **52.10/82.40** | **71.80** | **31.70/62.30** |
| Ours | VAE (R) | 78.81 | 47.27/52.03 | 65.53 | 31.30/65.13 | 71.09 | **22.58**/59.36 |
| | LAE (R) | 81.75 | 50.30/52.50 | 67.15 | 36.47/**72.07** | **71.30** | 21.76/**60.55** |
| | TAE (R) | **87.14** | **61.56**/61.68 | **67.81** | **36.48**/71.12 | 70.14 | 20.65/58.86 |
| | R-RNN (R) | 82.75 | 58.44/**67.30** | 63.53 | 25.48/49.78 | 58.42 | 11.10/36.05 |
| | C-RNN (C) | 75.99 | 36.66/54.68 | 64.00 | 32.00/61.51 | 60.01 | 15.68/46.16 |

Table 3: Frame-level AUC, RBDC and TBDC scores achieved on the human-related datasets HR-Avenue, HR-ShanghaiTech and HR-UBnormal. We compare the results of our unsupervised approaches against other pose-based approaches from PAD. Our regression- and classification-based models are marked with an (R) and (C), respectively. We highlight the best results per metric and within each category (PAD/Ours) with bold font.

| | Method | HR-Avenue | | HR-ShanghaiTech | | HR-UBnormal | |
|---|---|---|---|---|---|---|---|
| | | AUC | RBDC/TBDC | AUC | RBDC/TBDC | AUC | RBDC/TBDC |
| PAD | Morais et al. (2019) | 86.30 | -/- | 75.40 | -/- | - | -/- |
| | Flaborea et al. (2023) | **89.00** | -/- | 77.60 | -/- | **68.40** | -/- |
| | Hirschorn & Avidan (2023) | - | -/- | **87.40** | -/- | - | -/- |
| Ours | VAE (R) | 76.85 | 56.86/64.70 | 65.74 | 31.90/66.11 | **72.50** | **23.59**/59.80 |
| | LAE (R) | **83.91** | **61.47**/66.66 | 67.36 | 37.00/**72.90** | 72.33 | 23.30/**61.04** |
| | TAE (R) | 75.63 | 46.49/**71.38** | **68.23** | **37.59**/72.00 | 71.87 | 21.84/59.55 |
| | R-RNN (R) | 83.18 | 59.44/67.24 | 64.12 | 27.49/53.23 | 58.13 | 10.70/37.78 |
| | C-RNN (C) | 75.43 | 37.29/56.17 | 64.36 | 33.15/61.99 | 60.24 | 14.47/46.86 |

increased safety in video surveillance settings which are privacy-preserving, computationally light-weight and transferrable across the imaging spectrum. We identified two solution approaches within which we presented five baseline models in total. These models either belong to the family of regression- or classification-based learning techniques in DL and follow shallow network architecture designs. We enabled the implementation of both tracks leveraging two distinct data representations for the extracted human contours: a radii-based feature descriptor and Shape Contexts. Overall, we found that the linear AE models we employed perform best across all six benchmark datasets which we evaluated our approaches on. In comparison to prior art

within the domains of VAD and PAD, we conclude that the use of contours to detect human-related anomalies in video represents a novel and promising path worth exploring in future research.

One particular advantage we see contours to have over skeleton-based approaches, is the opportunity to expand our set of presented experiments to include other object categories as well. In this context, we hypothesize that even (contours of) rigid objects experience a visual deformation as they move across the scene and leave the development of more sophisticated multi-class, contour-based approaches for VAD for future research.

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
