# OpenReview forum: "Video Anomaly Detection with Contours - A Study"
_TMLR — Withdrawn by Authors_

### Review · Reviewer_u4r4 · 2025-02-03

**Summary Of Contributions:**

This paper tackles video anomaly detection from human contours. The authors extract the human contours using a customized algorithm. Then, two feature representations, a radii-based feature descriptor, and shape contexts are extracted. Four regression-based methods (VAE, LAE, TAE, R-RNN) and a classification-based method are compared.

**Audience:**

No

**Claims And Evidence:**

No

**Requested Changes:**

Conduct experiments on different combinations of features and regression and classification models.

**Strengths And Weaknesses:**

Strengths
- Unlike human pose, contour information can be used in targets other than humans.
- Authors conducted experiments using several methods.

Weakness
- Contours information is introduced to broad category of target. However, the experiment is mainly conducted on human anomaly. The performance of human anomaly detection is lower than compared methods. Thus, its effectiveness is unclear.

- Two features (Raddi-based feature descriptor and Shape Context ) are introduced to regression and classification based methods, respectively. It is not well motivated for the use of different features to these different methods.

- In current form of experiments, reader cannot obtain insights of each method, e.g, why classification and regression-based approach are compared, which features are effective.

---

### Review · Reviewer_YxdS · 2025-03-08

**Summary Of Contributions:**

This paper focuses on the task of video anomaly detection (VAD), which aims to identify abnormal events in video sequences. Specifically, it explores the use of contour information to enhance VAD performance. The key contributions of this work are as follows:

- Investigating the potential of contours as effective appearance abstractions of humans for pose-based VAD, leveraging regression and classification methods from deep learning.
- Analysing two data representation techniques for extracted contours.
- Demonstrating that contour-based representations provide enhanced object anonymisation while reducing computational complexity.
Extensive experiments are conducted on VAD benchmark datasets (e.g., CUHK Avenue, ShanghaiTech UBnormal) to assess performance and facilitate easy comparisons for future research.

**Audience:**

Yes

**Broader Impact Concerns:**

I don't think the paper have any ethical concerns.

**Claims And Evidence:**

No

**Requested Changes:**

- Revise the writing of the paper to improve clarity and coherence, ensuring that all statements are properly supported with appropriate citations.

- Include citations, evaluations, and discussions of prior works in the field to provide a more comprehensive contextual background.

- Clearly define the learning task and training objective in the methodology section to enhance understanding.

- Strengthen the justification for the necessity of the proposed contour-based research study.

- Provide additional results on another real-world dataset, such as UCF-Crime, to demonstrate the generalisability of the approach.

- Conduct a computational complexity evaluation in the experimental section to assess the efficiency of the proposed method.

- Include qualitative evaluations and discussions on failure cases to highlight potential limitations and areas for improvement.

**Strengths And Weaknesses:**

### Strengths
- The paper introduces the use of 2D contours as an alternative to skeleton-based representations for pose-based video anomaly detection (PAD). This novel approach could offer advantages in terms of generalisation to other object categories beyond humans.
- The preservation of increased object anonymization with reduced computational complexity.

### Weaknesses
- I found that the representation and writing of the manuscript need improvement, as some sections are difficult to follow. For example, in the literature review section, the limitations of previous methods are generally not mentioned. Additionally, the motivation behind the selected learning frameworks is not explicitly stated. Furthermore, some statements lack proper citations in the introduction section.

- The paper lacks a lot of prior works in the field of video anomaly detection (e.g., RTFM [a], BN-WVAD [b], UR-DMU [c], etc.). Please refer to the Papers with Code website to include, evaluate, and discuss these methods in the literature review and experimental sections.

- The problem and learning target are not mathematically defined in the methodology section, which may reduce the paper's readability.

- The evaluated baseline method are generally old methods in the deep learning field. I am wonderind if the author have considered to setup a baseline method that is based on the vision transformer architectual?

- I would appreciate it if the authors could further clarify the motivation for using object contours instead of semantic segmentation. What are the advantages of using contours? Could a semantic segmentation mask be used instead?

- Although the proposed the method show better results on the UBnormal benchmark, the model performance is significantly worse on CUHK and ShanghaiTech benchmark. It would be great to see to model performance on other complex dataset such as UCF-Crime.

- The author claimed that the contour-based method can reduce computational complexity. However, there are no specific experimental results to support this claim.

- Qualitative results are missing from the experimental section, limiting the exploration of the advantages and disadvantages of the proposed contour-based methods.

[a] Tian, Y., Pang, G., Chen, Y., Singh, R., Verjans, J.W. and Carneiro, G., 2021. Weakly-supervised video anomaly detection with robust temporal feature magnitude learning. In Proceedings of the IEEE/CVF international conference on computer vision (pp. 4975-4986).

[b] Zhou, Y., Qu, Y., Xu, X., Shen, F., Song, J. and Shen, H.T., 2024. Batchnorm-based weakly supervised video anomaly detection. IEEE Transactions on Circuits and Systems for Video Technology.

[c] Zhou, H., Yu, J. and Yang, W., 2023, June. Dual memory units with uncertainty regulation for weakly supervised video anomaly detection. In Proceedings of the AAAI Conference on Artificial Intelligence (Vol. 37, No. 3, pp. 3769-3777).

---

### Review · Reviewer_iPzv · 2025-03-10

**Summary Of Contributions:**

The paper presents an approach to video anomaly detection using contour as input. There are two contour extraction approaches that are used to extract contours and provide them to a set of different methods to perform anomaly detection. The paper examines several schemes for anomaly detection, namely a variational auto-encoder, a linear auto-encoder, a tabular auto-encoder, an RRN, and classification as a clustering approach. All methods are evaluated on six standard benchmarks. The results are mixed between the baselines evaluated.

**Audience:**

Yes

**Claims And Evidence:**

No

**Requested Changes:**

The paper needs a lot of work. There are no minor changes at the moment. The method needs further work and additional evaluations should be done.

Overall, the paper presents an interesting perspective, but it is not fully developed. The contours show potential and it may be a promising direction. However, the paper is incomplete in terms of methodology and evaluation. This leads to a lack of novelty and significant contribution.

**Strengths And Weaknesses:**

Advantages:
- The idea is easy to follow. The paper is well written and easy to understand.
- The experimental part shows the potential of using contours instead of body poses.

Disadvantages:

- The main limitation of the paper is its novelty and its contributions. The idea of contours is interesting but not fully developed to provide state-of-the-art performance.
- In addition, the paper presents several approaches based on contours. They all look like proposed approaches. At the moment it reads as if many approaches have been tested and the results are reported for each approach. The paper lacks a clear message that would come from a main method. At present there is no main method. It would add more value to the paper to have many baselines in the experiments and compare them with the main method of the paper (which does not exist at the moment).
- The results reported are interesting, but far from state-of-the-art. The paper does not report the best methods, e.g. "Self-Distilled Masked Auto-Encoders are Efficient Video Anomaly Detectors" (CVPR2024), MULDE (CVPR2024).
- Some standard experiments are missing, e.g. UCF-Crime and Ped2. More recent results should also have been reported for the existing benchmarks.
- The results are mixed as to which methods perform best. This is another point of providing an incomplete method.

---

### Review · Reviewer_Urnd · 2025-03-12

**Summary Of Contributions:**

The paper introduces a novel approach to pose-based video anomaly detection by leveraging object contours to identify potential anomalous behavior. Based on this concept, two main strategies are proposed: regression methods (e.g., VAE, LAE) and classification methods (e.g., SVM). The paper utilizes six benchmarks to demonstrate the effectiveness of the proposed contour-based approach.

**Audience:**

No

**Claims And Evidence:**

No

**Requested Changes:**

The proposed contour-based approach is interesting and has the potential to advance the field. However, the methods used to evaluate video anomaly detection with contours are unconvincing, as they appear outdated. I recommend incorporating more recent deep learning mechanisms (e.g., Transformers, Mamba) for evaluation, which would likely attract greater interest from the community.

**Strengths And Weaknesses:**

Strengths:
1). The paper is well-written and provides extensive demonstrations, making it easy to understand the proposed work.
2). The paper conducts comprehensive experiments using six datasets.

Weaknesses:
1). The contour extraction process utilises YOLOv7 to generate bounding boxes, which then guide SAM in producing segmentation masks. However, the paper refers to SAM as a "semantic segmentation module," which is inaccurate since SAM generates class-agnostic masks without semantic information. Additionally, both YOLOv7 and SAM are considered outdated. Why not explore YOLOv11 and SAM2 instead?

2). In the introduction, you state: "Additionally, opposed to skeletons, this approach leaves the opportunity to be expanded to other object categories open for future research." This is a strong claim, yet no supporting citation is provided. Furthermore, all experiments in the paper focus on human as objects. Could you clarify which objects you are referring to? For instance, animals also have skeletons [1].

3). Similar to point (1), I am also puzzled by the use of one-class SVM for anomaly detection in the "Classification Way". Given the availability of more advanced methods, why was this approach chosen?

4). The paper lacks a comparison with [2, 3] in the experiments, where [2] appears to achieve better results on ShanghaiTech.


[1]. DeepSkeleton: Learning Multi-task Scale-associated Deep Side Outputs for Object Skeleton Extraction in Natural Images
[2]. Self-Distilled Masked Auto-Encoders are Efficient Video Anomaly Detectors
[3]. Weakly-supervised video anomaly detection with robust temporal feature magnitude learning

---

### Note · Authors · 2025-03-25

**Comment:**

We thank everyone involved for the time and effort put into providing us with a scientific evaluation of our work, and for sharing their professional expertise. We are glad to hear that overall, the paper is perceived as well-written and easy to understand. We also appreciate the suggestions that will help us create an improved version of our work, namely:

- Experimenting with a combination of feature vectors from the regression and classification approach into a merged solution proposal.
- Conducting a thorough qualitative evaluation of failure cases in an attempt to shed light into the diffuse landscape of obtained results in which no model performs constantly best across all benchmark datasets.
- Providing scientific proof in terms of additional experiments which support the claim that this method can be extended to more object categories than just humans.
- Extending our section on prior art.

At the same time, we had submitted our work to TMLR because it promised the following (see https://jmlr.org/tmlr/index.html and https://jmlr.org/tmlr/acceptance-criteria.html):

- "TMLR emphasizes technical correctness over subjective significance, to ensure that we facilitate scientific discourse on topics that may not yet be accepted in mainstream venues but may be important in the future."
and
- "Crucially, it should not be used as a reason to reject work that isn't considered “significant” or “impactful” because it isn't achieving a new state-of-the-art on some benchmark. Nor should it form the basis for rejecting work on a method considered not “novel enough”, as novelty of the studied method is not a necessary criteria for acceptance. We explicitly avoid these terms (“significant”, “impactful”, “novel”), and focus instead on the notion of “interest”."

Nonetheless, two out of the four reviewers (50%) of our paper claim that the models we chose are 'outdated' believing that the approach won't be of great interest to the community because it does not implement 'more recent deep learning mechanisms'. Additionally, we read that only a method that reaches state-of-the-art performance can be truly novel, which we politely disagree with. While we can understand that the presented approaches do not appear sophisticated enough within the field of interest, we underline that our intention has been to share scientific insights into a new data representation perspective for VAD. To achieve this to a broad extent, we tackled the task as a classification and regression problem implementing various baseline models. We will do our utmost to make this clearer in a new version of the manuscript.

**Withdrawal Confirmation:**

I have read and agree with the venue's withdrawal policy on behalf of myself and my co-authors.